# Challenges and Considerations during In Vitro Production of Porcine Embryos

**DOI:** 10.3390/cells10102770

**Published:** 2021-10-15

**Authors:** Paula R. Chen, Bethany K. Redel, Karl C. Kerns, Lee D. Spate, Randall S. Prather

**Affiliations:** 1Division of Animal Sciences, University of Missouri, Columbia, MO 65211, USA; 2USDA-ARS, Plant Genetics Research Unit, Columbia, MO 65211, USA; 3Department of Animal Science, Iowa State University, Ames, IA 50011, USA; 4National Swine Resource and Research Center, University of Missouri, Columbia, MO 65211, USA

**Keywords:** porcine embryo culture, in vitro maturation, in vitro fertilization

## Abstract

Genetically modified pigs have become valuable tools for generating advances in animal agriculture and human medicine. Importantly, in vitro production and manipulation of embryos is an essential step in the process of creating porcine models. As the in vitro environment is still suboptimal, it is imperative to examine the porcine embryo culture system from several angles to identify methods for improvement. Understanding metabolic characteristics of porcine embryos and considering comparisons with other mammalian species is useful for optimizing culture media formulations. Furthermore, stressors arising from the environment and maternal or paternal factors must be taken into consideration to produce healthy embryos in vitro. In this review, we progress stepwise through in vitro oocyte maturation, fertilization, and embryo culture in pigs to assess the status of current culture systems and address points where improvements can be made.

## 1. Introduction

In vitro production of embryos has several advantages over in vivo-derived embryo production, including efficient selection of superior genetics for transfer or genetic modification to rapidly obtain animals with desirable traits. Genetic engineering provides a powerful tool to aid in understanding basic mechanisms regulating animal physiology. Several studies using genetic engineering approaches have been conducted to ascertain the function of signaling molecules and enzymes during early pregnancy in pigs [1,2,3,4]. Importantly, interleukin 1 beta 2 (IL1B2) was shown to be required for conceptus elongation to proceed [1]. For production agriculture, embryos can be modified to produce animals with better carcass traits, such as improved fatty acid profiles with increased levels of omega-3 fatty acids [5], and resistance to diseases, such as porcine reproductive and respiratory syndrome (PRRS), that result in millions of dollars of losses per year [6,7]. Moreover, genetically modified swine have become powerful tools for studying genetic diseases in humans, such as cystic fibrosis [8] and phenylketonuria [9], and for xenotransplantation [10]. Therefore, production and manipulation of porcine embryos in vitro is crucial for advancements in agriculture and human medicine.

Each step of the culture system, from oocyte maturation to embryo culture, has been continuously optimized and adjusted based on the requirements of porcine oocytes and embryos. Our current system, from collection of cumulus-oocyte complexes (COCs) to surgical transfer into surrogates, is outlined in Figure 1 with media formulations described in the subsequent sections. However, in vitro culture of porcine embryos is still suboptimal as only about 40% of presumptive fertilized oocytes develop to the blastocyst stage for embryo transfer [11], and the number of cells in the blastocyst-stage embryos is decreased compared to those that developed in vivo [12]. Clues from other species have provided insight for culture of porcine embryos, but embryos from every species exhibit unique metabolic characteristics that need to be considered for media formulations. Even with advances in porcine embryo culture media formulations, stress from the environment and maternal or paternal factors can impact the developmental trajectory of the embryos. Thus, a significant amount of information has yet to be learned regarding the interaction of embryos with the culture environment, the ways in which the culture system can be modified to improve developmental viability, and important metabolic features that can be used to distinguish healthy embryos.

Herein, we provide an overview of the current porcine embryo culture system with a focus on the metabolic characteristics of porcine embryos. Moreover, measurements of embryo quality, stressors arising from the culture system, and parental factors that influence development are discussed.

## 2. Oocyte Quality and Maturation

### 2.1. Selection Criteria

In vitro maturation (IVM) of oocytes is the first critical step in the process of producing competent embryos that are able to produce viable offspring. While progress has been made to improve IVM of pig oocytes, the rate and the extent of maturation is considerably lower than in vivo-matured oocytes, suggesting a suboptimal maturation environment [13]. In vivo, the maternal ovarian follicle environment supplies the immature oocyte with the necessary components to develop and reach its developmental competence by maintaining an intimate relationship with its neighboring somatic cells. The bidirectional communication between the oocyte and cumulus cells allows for the oocyte to gradually acquire the necessary molecular and cytoplasmic machinery needed to support embryo development [14,15]. 

Evidence suggests that the key factor in determining the proportion of oocytes that develop to the blastocyst stage appears to be the intrinsic quality of the oocyte that begins the in vitro production process [16]. Given this, the ability to select the most competent oocytes that are capable of being fertilized in vitro and produce healthy young is imperative. General selection parameters to determine oocyte competence in the pig are the age of the oocyte donor [17,18], follicle size [19], and the layers of cumulus cells and subsequent cumulus cell expansion during culture [20]. Oocytes that originate from small (<3 mm) follicles are less likely to mature to metaphase II (MII) or develop to the blastocyst stage compared to oocytes derived from medium (3–5 mm) and large (>5 mm) follicles [21]. Moreover, larger follicles (>5 mm) have been shown to have higher β-estradiol concentrations, and high quality COCs from these follicles have increased abundance of transcripts involved in ovarian steroidogenesis [22]. Historically, cumulus cell expansion has been considered to be required for successful maturation in vitro and often the degree of cumulus cell expansion is correlated with improved oocyte maturation [23,24]. As another factor to consider, oocyte quality is also impacted by the energy status of the female as feed restriction (50% per day) during the last two weeks of lactation decreased follicle size, concentration of steroids in the follicular fluid, and maturation to MII [22]. 

### 2.2. Supplementation of Growth Factors

A cocktail of growth factors, fibroblast growth factor 2 (FGF2), leukemia inhibitory factor (LIF), and insulin-like growth factor 1 (IGF1) (termed FLI) supplemented only during oocyte maturation of prepubertal gilt COCs, was found to provide a four-fold increase in the number of piglets born per oocyte collected [11]. These FLI-cultured COCs had a distinct mitogen-activated protein kinase (MAPK) activation pattern in the cumulus cells, displayed a higher degree of cumulus cell expansion, and accelerated the disruption of gap junctions compared to control COCs. However, a complementary study found that in the absence of gonadotropins but with supplementation of FLI, cumulus cells showed little expansion but maturation to MII was not impeded in oocytes from prepubertal gilts, and those oocytes were competent to produce healthy piglets [25]. While cumulus cell expansion was not needed to produce competent oocytes, cumulus cells are required as their removal at the beginning of maturation and even up to 24 h post-start of maturation impedes oocyte development to MII and the blastocyst stage after fertilization. Our knowledge and understanding about the relationship of the oocyte, cumulus cells, growth factors, and hormones are constantly evolving as a highly complex interaction is revealed. More research is needed to understand the molecular mechanisms and signaling events by which FLI can promote oocyte competence.

## 3. In Vitro Fertilization

Historically, the lack of fertilization of porcine oocytes in vitro [26] was followed by tremendous polyspermy [27]. The inability to achieve consistent monospermy has been a major problem for in vitro fertilization (IVF) of the pig. In the mid-1990s, Billy Day from the University of Missouri was quoted as saying, “For years, we tried to get the sperm in, and now, we can’t keep them out.” Biologically, the block to polyspermy in mammals is via two distinct mechanisms. The first is a membrane block that acts relatively quickly [28], and the second is a block at the level of the zona pellucida that takes minutes. The relative importance of these two mechanisms is different between mammalian species. In pigs, humans, and mice, both mechanisms are used. The first relatively quick membrane block appears to act by changing the binding characteristic of the plasma membrane of the oocyte, whereas the second zona pellucida block is a result of the release of cortical granule contents into the space between the oolemma and the zona pellucida. These contents (e.g., ovastacin [29], among others [30]) modify the proteins of the zona pellucida such that sperm penetration is inhibited. Thus, these two mechanisms serve to complement each other in preventing multiple sperm from entering the oocyte cytoplasm.

In the pig, the block at the zona pellucida is relatively strong and is correlated with cortical granule release [31], while the membrane block is regulated by calreticulin [32]. Many efforts have been made to alter the fertilization conditions to improve monospermy rates. These efforts have included altering the concentration of sperm while varying the duration of exposure between the sperm and oocytes, adding follicular fluid [33] or substances found in oviductal fluid, such as deleted in malignant brain tumors 1 (DMBT1) [34], oviductal glycoprotein 1 (OVGP1) [35], secreted phosphoprotein 1 (SPP1) [36], or plasminogen [37,38], as well as trying to recreate the oviductal environment by adding oviductal fluid [39,40] or oviductal extracellular vesicles [41], among other things [42].

The problem of polyspermy encountered during IVF is generally confounded with in vitro-matured oocytes. Improvements in the quality of the oocytes will likely improve the success of monospermic fertilization. As in vitro-matured oocytes have a propensity to retain their transzonal projections, it was thought that these projections may interfere with sperm-oocyte binding and cortical granule exocytosis [43]. Retraction of those projections at the end of oocyte maturation may permit more timely and complete cortical granule exocytosis (the slower block to polyspermy), as well as altering the composition of the plasma membrane to provide a better fast block to polyspermy at the level of the plasma membrane. The aforementioned maturation system containing FLI has dramatically altered the timing of the retraction of the transzonal projections and improved the overall developmental competence of the oocytes [11]. As compared to other reports in the literature, polyspermy is less of a problem in our hands. A fertilization rate of 50–60% with monospermy of 70–80% is routinely achieved. Thus, on a per oocyte basis, the overall success is 35–50% monospermy. This range of monospermy corresponds well with our rates of development to the blastocyst stage (30–60%).

Another alternative to counteract polyspermy is to perform intracytoplasmic sperm injection (ICSI), which has led to the production of live piglets [44,45]. However, delays in cleavage divisions and decreased blastocyst formation of approximately 10–20% have been observed in ICSI-derived porcine embryos compared to those produced by IVF [46]. These decreases in development may be due to selection of damaged sperm and/or incomplete oocyte activation. Importantly, boar sperm selected by a Percoll gradient were shown to have higher abundance of phospholipase C-ζ (PLCζ), a well-studied sperm-borne oocyte activation factor, and the use of Percoll-selected sperm for ICSI increased monospermic fertilization and oocyte activation as well as development to the blastocyst stage [47]. Several protocols use polyvinylpyrrolidone (PVP) for sperm immobilization before ICSI; however, this compound is toxic when injected into oocytes. When boar sperm were selected by hyaluronic acid (HA), known as physiological intracytoplasmic sperm injection (PICSI), versus PVP (conventional ICSI) or IVF, increases in development to the blastocyst stage and total cell numbers were observed [48].

## 4. Comparing Metabolic Characteristics of Mammalian Embryos 

### 4.1. Carbohydrates

The metabolism of preimplantation embryos differs drastically from that of adult somatic cells. In normal conditions, somatic cells metabolize glucose through glycolysis, yielding two ATP, two NADH, and two pyruvate molecules. Pyruvate is further metabolized through the TCA cycle to produce electron donors, NADH and FADH_2_, for generation of 30–36 ATP molecules by oxidative phosphorylation. In the precompaction embryo, ATP is generated by oxidative phosphorylation at a reduced but optimized level, which is sustained through oxidation of pyruvate, amino acids, and fatty acids [49]. A low level of pyruvate oxidation is deemed ideal to avoid excess production of reactive oxygen species (ROS) by the electron transport chain. After compaction, the primary metabolic pathway is glycolysis that is characterized by the Warburg effect (WE), which is a phenomenon first observed in cancer cells by Otto Warburg [50,51]. Under the WE, glycolytic intermediates are shuttled towards the pentose phosphate pathway (PPP) and lactate production instead of the TCA cycle even in the presence of oxygen. The PPP supports rapid proliferation of cells by producing ribose-5-phophate, a precursor for nucleotides, and NADPH, which is involved in redox regulation and lipid synthesis [52,53]. Recent studies have shown that glucose does not drive ATP generation during most stages of preimplantation development, but that it is metabolized through the PPP and the hexosamine biosynthesis pathway to control localization of transcription factors, yes-associated protein 1 (YAP1) and transcription factor AP-2 gamma (TFAP2C), for cell fate specification [54]. Thus, generation of biomass, such as DNA and membrane lipids, and orchestration of key events during development are potentially the main roles of glucose to support rapid proliferation during these early developmental stages. 

Classically, pyruvate and lactate are known to be important carbohydrates during early stages of preimplantation development, and glucose becomes a key energy source after compaction [55,56,57]. Pig embryos have decreased metabolism of pyruvate compared to mouse and sheep embryos, pointing to a reliance on other energy sources, such as glucose or lipids [58,59,60,61]. Studies in several species, including mouse, hamster, sheep, cow, and human, have reported that glucose inhibits the development of embryos in vitro [62,63,64,65,66]. Glucose does not appear to be inhibitory for porcine preimplantation embryo development in vitro, but glucose in the presence of inorganic phosphate has been shown to be deleterious [67,68]. In porcine embryos, glucose metabolism minimally increases from the 1-cell to 8-cell stage (0.4 ± 0.1 to 3.1 ± 0.4 pmol/embryo/4 h) but notably increases in the compacted morula (49.5 ± 2.7 pmol/embryo/4 h) and further by the blastocyst stage (147 ± 12.0 pmol/embryo/4 h) [69]. Subsequent studies have also shown that glucose metabolism increases after compaction with glycolysis becoming a key metabolic pathway, but metabolic activity is significantly decreased in in vitro-produced embryos compared to in vivo counterparts [56,61]. However, porcine embryos can be cultured to the blastocyst stage in medium with no glucose but containing only pyruvate and lactate [70]. 

### 4.2. Amino Acids

Amino acids serve numerous functions in promoting the development of preimplantation embryos. Oviductal and uterine fluids contain varying concentrations of amino acids, which has influenced media formulations for several species [71,72,73]. Aside from protein synthesis, amino acids are involved in other activities, such as ATP production, nucleotide synthesis, lipid synthesis, antioxidant production, cell signaling, osmolarity, and pH regulation. Uptake of amino acids from the microenvironment by preimplantation embryos relies on the activity of different transport systems, which are expressed during different stages [74]. For example, in porcine oocytes, Na^+^-dependent transport of L-alanine and L-leucine was shown to be competitive and occurred by using the B^0,+^ system [75]. At the blastocyst stage, L-leucine was transported in a Na^+^-dependent manner through the L system. Transport of L-leucine into the embryo has been shown to be involved in activation of mechanistic target of rapamycin complex 1 (MTORC1) to promote trophoblast motility [76]. In support of this, treatment of mouse blastocyst-stage embryos with 200 nM rapamycin, an inhibitor of MTORC1, blocked outgrowth formation in vitro [74]. Sodium-dependent transport of L-aspartate and L-glutamate was not detected in porcine oocytes but was observed at the blastocyst stage, demonstrating that the X^-^_AG_ system becomes active at later stages in development [77].

The addition of essential amino acids to embryo culture media has been shown to be inhibitory for development to the blastocyst stage in mouse and cattle embryos [78,79]. However, Steeves and Gardner [80] demonstrated that presence of essential amino acids in medium with nonessential amino acids and glutamine was not inhibitory during cattle embryo cleavage and increased development to the blastocyst stage. In porcine parthenogenetic embryos cultured in modified Whitten’s medium with polyvinyl alcohol (PVA), essential amino acids were inhibitory if present at the 1-cell stage but supported development to the blastocyst stage when added after 48 h of culture. Specifically, presence of nonpolar essential amino acids, valine, leucine, isoleucine, and methionine, during the first 48 h of culture was shown to inhibit development past the 4-cell stage [81]. However, the porcine zygote medium (PZM) variants, containing both essential and nonessential amino acids, have been shown to support development of porcine embryos to the blastocyst stage [70]. 

Furthermore, transcriptional profiling of porcine blastocyst-stage embryos revealed that transcripts related to amino acid transport and metabolism were dysregulated in those cultured in vitro versus those derived in vivo [12]. In vitro-produced blastocyst-stage embryos had increased abundance of solute carrier 7A1 (*SLC7A1*), which encodes an arginine transporter, and supplementation of 1.69 mM arginine to PZM-3 decreased abundance of *SLC7A1* [82]. Arginine is metabolized for production of nitric oxide, and addition of nitric oxide synthase inhibitors to culture media has been shown to inhibit development of mouse and pig embryos [82,83]. Similarly, in vitro-produced porcine embryos had increased abundance of solute carrier 6A9 (*SLC6A9*), which encodes a glycine transporter, and supplementation of 10 mM glycine decreased *SLC6A9* abundance and improved development of blastocyst-stage embryos [84]. However, transfer of blastocyst-stage embryos cultured with 10 mM glycine did not result in pregnancies, unlike controls cultured with 0.1 mM glycine, showing that more cells at the blastocyst stage is not necessarily an indicator of developmental competence. Furthermore, early mouse and hamster embryo culture media formulations demonstrated the importance glutamine during cleavage stages to overcome the 2-cell block and for the development of blastocyst-stage embryos [62,85]. Glutamine has been shown to support development of porcine embryos from the 1- to 2-cell stage to the blastocyst stage in vitro without the presence of glucose [68]. Recently, glutamine supplementation to PZM-3 has been shown to correct abundance of transcripts related to glutamine transport and metabolism and increase activation of mTORC1 in porcine blastocyst-stage embryos [86,87]. Thus, RNA-sequencing is a powerful tool that can identify amino acid requirements of the embryo in culture, but viability in vivo must be assessed before supplementation of specific amino acids becomes common practice. 

### 4.3. Lipids 

In addition to carbohydrates, beta-oxidation of fatty acids is thought to contribute to sustaining oxidative phosphorylation in preimplantation embryos. Sturmey and Leese [60] demonstrated that oxygen consumption and ATP production increase at the early blastocyst stage, and ATP is mainly produced via aerobic metabolism. Treatment of mouse embryos with inhibitors of beta-oxidation, methyl palmoxirate and etomoxir, decreased the number of embryos reaching the blastocyst stage and total cell numbers within the embryos [88,89]. Interestingly, the inhibitory effects of etomoxir were decreased when porcine oocytes were matured in the presence of high glucose (4.0 mM) compared to low glucose (1.5 mM), thus other energy sources can compensate when beta-oxidation is blocked [90]. On the contrary, addition of L-carnitine, which is essential for uptake of fatty acids into the mitochondria, to embryo culture medium improved mitochondrial activity and development in embryos of several species [88,91,92,93]. 

Porcine oocytes and embryos have a dark appearance compared to other species due to presence of abundant lipid. Mouse oocytes were determined to have approximately 4 ng of fatty acids; sheep and cattle oocytes have a slightly darker appearance and about 89 and 63 ng of fatty acids, respectively; pig oocytes, almost black in appearance, have about 156 ng of fatty acids [49,94,95]. The most common fatty acids in pig oocytes are palmitic (16:0), stearic (18:0), and oleic (18:1n-9) acids [95]. Compared to other species, the large amount of lipid in pig oocytes is potentially the result of increased message for diacylglycerol acyltransferase 1 (*DGAT1*), which encodes an enzyme involved in the transfer of a fatty acyl CoA to a diacylglycerol to form triglycerides [96]. Moreover, inhibition of stearoyl-coenzyme A desaturase 1 (SCD1), which introduces a cis double bond in saturated fatty acids, decreased development to the blastocyst stage and lipid droplet formation in porcine parthenogenetic embryos; however, cotreatment with 100 μM oleic acid rescued development [97]. Sturmey and Leese [60] observed that triglyceride levels did not change from cleavage stages to the blastocyst stage, indicating that beta-oxidation might not be a key metabolic pathway in the presence of other energy sources. However, Romek et al. [98] noted that the volume of lipid droplets per unit of cytoplasm decreased at the blastocyst stage in both in vitro-produced and in vivo-derived porcine embryos, demonstrating controversy in this area. 

The large amount of lipid in porcine oocytes and embryos has been identified as a barrier to vitrification and survival after thawing. In addition, porcine oocytes have a greater percentage of polyunsaturated fatty acids compared to ruminants, which further impairs survival after freezing [95]. L-carnitine supplementation, centrifugation in a high-osmolarity solution to separate lipids from the cytoplasm, and removal of lipids by micromanipulation have been shown to be effective methods of increasing the cryotolerance of porcine oocytes and embryos [99,100,101,102].

## 5. Strategies for Improving Current Culture Systems

### 5.1. Media Formulations and Supplements

Several media formulations exist that support development of porcine zygotes to the blastocyst stage, including modified Whitten’s medium [103], North Carolina State University (NCSU)-23 medium [104], modified CZB medium [105], Beltsville embryo culture medium (BECM)-3 [106], and porcine zygote medium (PZM) variants [70]. Currently, NCSU-23 and PZM variants are most commonly used for porcine embryo culture. NSCU-23 contains glucose and glutamine as primary energy sources as well as taurine and hypotaurine to mediate osmolarity but lacks pyruvate, lactate, and all other amino acids [104]. PZM variants were formulated based on the composition of porcine oviductal fluid, containing pyruvate and lactate instead of glucose, and they lack taurine but include all other amino acids. Furthermore, PZM-3 contains bovine serum albumin (BSA), similar to NCSU-23, but culturing IVF or somatic cell nuclear transfer (SCNT)-derived embryos in PZM-3 was shown to improve development to the blastocyst stage and increase the number of cells in blastocyst-stage embryos compared to NCSU-23 [70,107]. PZM-4 substitutes PVA for BSA in an attempt to create a chemically defined medium [70]. Bovine serum albumin can contain undefined components, such as citrate and lipids, and different lots need to be tested before being routinely added to culture media. When embryos were cultured in PZM-4 compared to PZM-3, no difference in development was detected, and live piglets were born after embryo transfer [70]. However, subsequent studies demonstrated decreases in development to the blastocyst stage by using PZM-4 compared to PZM-3 [108]. 

Additives to PZM-3, such as amino acids and small molecules, have been shown to further improve porcine embryo development. For example, supplemental concentrations of arginine increased development to the blastocyst stage, improved embryo quality, and modulated transcript abundance related to arginine transport, resulting in a new medium named MU1 [82]. MU2 was developed after addition of 5-(4-Chloro-phenyl)-3-phenyl-pent-2-enoic acid (PS48) was shown to be able to replace BSA and improved development through stimulation of phosphoinositide 3 kinase (PI3K) to increase phosphorylation of v-akt murine thymoma viral oncogene homolog (AKT) [109]. Afterwards, MU3 was developed by replacing 1 mM glutamine with 3.75 mM GlutaMAX, an L-alanyl-L-glutamine dipeptide, which improved development to the blastocyst stage and increased mitochondrial activity [86]. Currently, MU4 has been implemented, which does not contain hypotaurine as removal of this component did not have a negative effect when embryo culture is conducted at 5% O_2_ [110], and hypotaurine is a relatively expensive ingredient.

### 5.2. Morphological and Chromosomal Quality

Embryos developing in vivo are exposed to the oviductal and uterine epithelia, which are involved in provision of nutrients, removal of toxins, and production of antioxidant systems [111]. Embryos of several species fertilized and cultured in vitro demonstrate a delay of about one cell division to the blastocyst stage compared to in vivo-derived embryos [12,112,113,114]. Total cell numbers in porcine blastocyst-stage embryos produced in vitro are decreased and have higher ratios of trophectoderm to inner cell mass [12,112]. Conventionally, morphology of an embryo has been the main criterion for determining quality [115]. However, morphology should not necessarily be the primary selection standard as porcine embryos cultured with supplemental glycine exhibited improved development and cell numbers but failed to establish pregnancies after 11 embryo transfers [84]. When the synchronization protocol was altered to transfer embryos cultured with supplemental glycine on an earlier day of standing estrus (typically performed on day 3, 4, or 5 of standing estrus), one pregnancy was obtained (unpublished data), indicating that these embryos may have increased developmental competence. Similarly, addition of N-methyl-D-aspartic acid (NMDA) and homocysteine (HC) to PZM-4 increased the percentage of porcine embryos that developed to the blastocyst stage and sizes of the embryos over PZM-4 alone, but 16 transfers of embryos cultured with only NMDA or NMDA and HC did not result in any live piglets [108]. The use of time lapse monitoring of development in vitro has been shown to increase pregnancy rates over conventional morphological assessment [116]. This technology allows for the consideration of other factors, such as timing of first cleavage and number of blastomeres after the first cleavage, which have both been shown to be more reliable indicators of quality [117,118]. The ability of porcine embryos to reach the morula stage before 102 h was correlated with increases in reaching the blastocyst stage, and the percentage of fragmentation negatively correlated with developmental progression [119].

Furthermore, chromosomal abnormalities can occur during development that significantly increase the chances of pregnancy loss or defects after birth. By using comparative genomic hybridization, approximately 14% of in vivo-derived porcine embryos (72 h after insemination) were observed to be aneuploid [120]. However, about 39% of in vitro-produced porcine blastocyst-stage embryos demonstrated chromosomal abnormalities [121]. Assessment of chromosomes 6 and 7 in in vitro-produced bovine blastocyst-stage embryos revealed that 72% of the embryos contained cells that were polyploid; however, the proportion of abnormal cells within each embryo was generally low (<10% of the total cells) [122]. More recent studies have observed high occurrence chromosomal aberrations in at least one blastomere of IVM-IVF bovine embryos as compared to in vivo-derived embryos (85% vs. 19%) [123] and that entire parental genomes can segregate into different cell lineages during cleavage divisions [124]. 

### 5.3. Mitochondrial Function

Mitochondria are dynamic organelles with important roles during fertilization and preimplantation development. From the zygote to expanded blastocyst stage, mitochondrial morphology changes from spherical with minimal cristae to elongated with numerous transverse cristae, demonstrating a major shift in metabolic activity during preimplantation development [125]. Mitochondrial membrane potential (ΔΨm) is formed by the pumping of protons across the inner membrane during oxidative phosphorylation and is commonly used as an indirect measure of mitochondrial function and developmental progression in preimplantation embryos. For example, JC-1 is a cationic, lipophilic dye that remains in its monomeric form at low (<100 mV) ΔΨm and fluoresces green but forms aggregates inside mitochondria at high (>140 mV) ΔΨm and fluoresces red [126]. Thus, the ratio of red to green fluorescence is an indicator of mitochondrial membrane potential and hence activity within an embryo. JC-10, which has better water solubility than JC-1, can effectively visualize mitochondrial activity in porcine blastocyst-stage embryos derived by IVF and SCNT [86,127], and glutamine supplementation into the culture medium increased the mitochondrial activity of the IVF-derived embryos [78]. Additionally, MitoTracker™ Green is non-fluorescent in aqueous solutions but accumulates in the mitochondrial matrix regardless of ΔΨm and fluoresces green to measure total mitochondrial mass. MitoTracker™ Red or Orange are oxidized by molecular oxygen in mitochondria to emit a red-orange fluorescence that can be used to measure oxidative activity [128]. MitoTracker™ Red has been used to determine mitochondrial activity of porcine IVF-derived blastocyst-stage embryos after culture with supplemental glycine; however, no difference in mitochondrial activity was observed [84].

### 5.4. Transcriptional Profiling

Transcriptional profiling has been used to elucidate “the needs” of the embryo. Studies using serial analysis of gene expression (SAGE) and microarrays revealed differences in abundance for transcripts related to cellular metabolism and transcriptional regulation in in vitro-produced porcine blastocyst-stage embryos compared to in vivo-derived counterparts [129,130]. By using next-generation sequencing (NGS), Bauer et al. [12] identified 1170 differentially abundant transcripts between in vivo-derived and in vitro-produced porcine blastocyst-stage embryos. Analysis of these data revealed that amino acid transport and metabolism were perturbed in in vitro-produced embryos. Specifically, message for an arginine transporter, *SLC7A1*, and a glycine transporter, *SLC6A9*, were upregulated in in vitro-produced embryos by 63- and 25-fold, respectively. As mentioned previously, supplementation of arginine to 1.69 mM increased development to the blastocyst stage and decreased abundance of *SLC7A1* to levels observed in in vivo-derived embryos [82]. Likewise, glycine supplementation to 10 mM improved developmental parameters of the embryos and decreased abundance of *SLC6A9*; however, pregnancies were not established after 11 transfers with embryos cultured in supplemental glycine [84]. Additionally, supplementation of glutamine improved development of porcine embryos and corrected abundance of transcripts related to glutamine transport and metabolism [78]. The use of RNA-sequencing datasets for defining and improving embryo culture is still in its infancy stages as there are numerous pathways yet to explore for improving media formulations. 

### 5.5. Metabolomics

Consumption or production of metabolites from the culture medium has been used as a noninvasive marker of embryo viability. Metabolomics assays are largely applied in IVF clinics to select embryos for transfer; however, these techniques are also useful for selection of embryos from agricultural species for cryopreservation or transfer. Metabolism of glucose, pyruvate, and lactate has been a canonical measure of preimplantation embryo competence, which was studied by using ultramicrofluorescence assays that measure NADH oxidation to represent pyruvate conversion to lactate or NADPH formation to represent glucose uptake [131]. Hardy et al. [132] observed that increased pyruvate consumption by early cleavage-stage human embryos correlated with increased development to the blastocyst stage. Regarding in vitro-produced porcine embryos, metabolism of glucose for glycolysis and pyruvate for the TCA cycle increased after compaction, indicating greater metabolic activity [61]. Glutamine and arginine have been shown to be consumed from culture media by human and pig embryos [133,134]. However, increased arginine consumption was associated with increased development to the blastocyst stage in pig embryos [133] but was associated with decreased development to the blastocyst stage in human embryos [134].

Currently, different platforms are being used to study metabolomic profiles of preimplantation embryos. Gas or liquid chromatography (GC or LC) coupled with mass spectrometry (MS) are ideal for analyzing metabolites in small volumes of culture media (5–10 μL). After derivatization, the mass of compounds allows for the identification of metabolites based on database information [52]. This technique has been successfully applied to porcine embryos to determine the production and consumption of metabolites from the medium after glutamine supplementation [86]. Culturing embryos with supplemental glutamine increased leucine consumption from the medium as well as activation of mTORC1 [78,79]. Additionally, nuclear magnetic resonance (NMR) spectroscopy has been used to analyze metabolites in spent culture media; however, this technique requires larger sample volumes (25 μL) [135,136].

### 5.6. Microfluidics 

Changing the type of culture system may provide methods to better customize the environment surrounding the developing embryos. Most current systems are static in a single atmosphere. The ability to continuously modulate the composition of the culture medium as the needs of the embryo change may provide dramatic improvements in developmental competence. Microfluidic devices are in development, and there are reports in the literature specific to pigs. Such devices have been reported to reduce osmotic stress [137], remove the zona pellucida [138], facilitate fertilization and reduce polyspermy [139,140], and customize the culture environment to improve development [141,142]. Moreover, microfluidic devices may improve tolerance of embryos to cryopreservation as addition and removal of cryoprotectants inflicts osmotic shock that may be detrimental to development. Use of a microfluidic device to gradually change the osmotic pressure during cryoprotectant addition or removal may increase the cryosurvival of pig oocytes or embryos [137]. 

### 5.7. Extended Culture

By day 12 of gestation, porcine embryos have undergone several changes in morphology, including spherical, ovoid, tubular, and filamentous forms, transitioning from tubular to approximately 100 mm in length in 1 to 2 h. Methods of culturing porcine embryos past the blastocyst stage as a measure of competence have been met with challenges that have hindered progress in this area. Spherical embryos encapsulated in double-layered alginate beads were able to attain a tubular form with increased estradiol-17β production in the culture medium [143]. However, the embryos did not elongate into filamentous forms; thus, maternally secreted factors and extracellular matrix components may be required to initiate this process. Chemical modifications to the alginate hydrogels, such as covalent attachment of RGD peptides or incorporation of 0.1 μg/mL SPP1, increased survival and promoted morphological changes [144]. Nonetheless, full elongation of porcine embryos in vitro has not been achieved and will require further investigation.

### 5.8. Cryopreservation

In many species, cryopreservation of sperm, embryos, and somatic cells is straightforward. Cryopreservation of pig somatic cells, such as fibroblast cells, can be achieved with conventional freezing systems used for other species. In contrast, pig sperm and embryos present some unique challenges. In all likelihood, the cell type with most variable survivability is boar sperm. Variation in viability is season-, breed-, boar- and ejaculate-specific. Commercial application of artificial insemination has historically been by using fresh semen. Since pigs are litter bearing, considerable genetic selection pressure can be placed on the offspring resulting from artificial insemination. As genetic progress can be made by using fresh semen, the lowered success rates from cryopreserved semen does not justify the lower farrowing rate (~20–30%) and decrease in litter size as compared to fresh semen [145]. 

While cryopreservation of semen presents difficulties, embryos are even more challenging. It is thought that the high lipid content of pig embryos presents the main obstacle to successful cryopreservation. To circumvent this problem, Nagashima [102,146] developed a method to centrifuge the early embryo, thus stratifying the lipids within the zona pellucida. Then, the lipids could be removed via micromanipulation, and the resulting embryos were cryopreserved by using conventional methods. One would reason that the lipids were necessary for development of the embryo; however, in at least one report, development was enhanced after removal [99]. Lipid removal by micromanipulation is labor-intensive. High speed centrifugation to completely separate the lipids within the zona pellucida has been used as a high throughput lipid removal technique for cryopreservation of in vitro-produced embryos [147]. Solid surface vitrification of oocytes or zygotes may be another alternative as 15 piglets were derived from 8 embryo transfers [148]. 

Although few reports on conventional cryopreservation of porcine embryos exist, those reports of successful cryopreservation of early pig embryos have not been widely repeatable, and the industry has not adopted the transfer of frozen embryos as a method of improving genetics or moving genetics around the world. As expected, in vitro-derived embryos are less viable than in vivo-derived embryos, and thus do not survive the rigors of cryopreservation to the same extent as in vivo-derived embryos. Continual efforts are being made to improve the quality of oocytes matured and embryos cultured in vitro. Improvements, such as adding FLI to the oocyte maturation system, may result in increases in cryosurvival in the pig as has been observed in cattle [149]. Two recent reports describe the production of piglets after cryopreservation and embryo transfer. After transferring 553 embryos to 35 surrogates, 59 piglets from 14 litters were produced (59/553 = 10%) [150]. In another study, 180 embryos were transferred to 12 surrogates, producing 37 piglets from 8 sows (37/180 = 21%) [151]. Unfortunately, the number of embryos cryopreserved is not well described in either case. Thus, the percentages listed above may not fully describe the success of cryopreservation as it may be concluded that more embryos were cryopreserved than were transferred. The question of viability is then raised, i.e., were only the embryos of good quality cryopreserved and transferred? If so, then the percentages above may represent an overestimation of the true success of these technologies.

## 6. Environmental Stressors Arising from the Culture System

### 6.1. Oxygen Tension and Reactive Oxygen Species

During oxidative phosphorylation, ROS, such as superoxides, form as a result of premature leaking of electrons from the electron transport chain to molecular oxygen [152]. Therefore, mitochondria are the main source of ROS within cells, which can act as signaling molecules for differentiation or cause membrane, protein, and DNA damage at higher concentrations, potentially leading to apoptosis. Interestingly, Dalvit et al. [153] observed a significant increase in ROS concentrations from the oocyte at MII to the two-cell stage in bovine. ROS levels continued to increase until the late morula stage but decreased significantly during blastocyst formation. In vitro-produced embryos are more susceptible to ROS formation and damage compared to in vivo-derived embryos because of external stressors, such as fluctuating oxygen tension, exposure to visible light, media contaminants, and absence of maternal antioxidant systems [111]. Supplementation of antioxidants to porcine embryo culture media has been shown to improve development and reduce ROS [154,155], but these studies were conducted by using atmospheric (21%) O_2_ during the culture period which promotes ROS formation.

Oxygen tension during the culture period has an impact on subsequent embryonic development (Figure 2). One- and two-cell stage porcine embryos cultured in 5% O_2_ and 5% CO_2_ developed to the blastocyst stage at a rate of 50%, whereas culture in 2% or 21% O_2_ and 5% CO_2_ resulted in less than 10% development to the blastocyst stage [156]. Culture in 5% O_2_ instead of atmospheric (21%) O_2_ shifted the global patterns of gene expression in mouse embryos to be more similar to those derived in vivo [157]. Additionally, amino acid turnover, as an indicator of viability, by human embryos was lower when cultured at 5% O_2_ compared to 21% O_2_ [158]. Culture in low (5%) O_2_ increased cell numbers in in vitro-produced porcine blastocyst-stage embryos and the abundance of transaldolase 1 (*TALDO1*) and pyruvate dehydrogenase kinase 1 (*PDK1*), which are signatures of a Warburg Effect-like metabolism [51]. As mentioned previously, lowering the oxygen tension to 5% O_2_ eliminated the need to add hypotaurine, which functions as an antioxidant, to porcine embryo culture medium as removal of this component did not impair development nor increase apoptosis [110].

### 6.2. Temperature

Most porcine in vitro culture systems use a physiological temperature of 38.5 °C for the incubation (Figure 2). However, several studies have investigated the effects of increased temperature, or heat stress, on oocyte maturation and development in vitro. Heat stress during in vitro maturation may model the effects of summer months or fever in sows on oocyte or embryo quality. Porcine COCs exposed to heat stress (41.5 °C) for 24 h had decreased cumulus expansion, progression to MII, and development after fertilization [159]. Supplementation of antioxidants during maturation, including astaxanthin, melatonin, and resveratrol, has been shown to mitigate effects of heat stress and decrease apoptosis [160,161,162]. 

Increasing the incubator temperature above 43 °C during the embryo culture period has been shown to decrease development of porcine embryos in a temporal manner, but abundance of heat shock protein 70 was not increased after heat stress exposure [163]. When IVF-derived porcine embryos were exposed to 42 °C for 9 h after the late 1-cell stage, decreases in development to the blastocyst stage and increases in apoptotic nuclei were observed [164]. However, heat shock at the same conditions directly after fertilization increased the rate of cleavage and tended to increase development to the blastocyst stage compared to the non-heat shock controls [165]. Similarly, increases in development of parthenogenetic embryos was observed when heat shock occurred directly after oocyte activation [165], and subsequent analyses revealed that dephosphorylation of MAPK was increased in the heat-shocked embryos compared to controls [164].

### 6.3. Osmolality

The osmolalities of porcine oviductal and uterine fluids range from 290 to 320 mOsm during the first five days of gestation [73]. The osmolalities of the most commonly used culture media, PZM-3 and NCSU-23, are 288 and 291 mOsm, respectively [70]. Li et al. [73] demonstrated that culture of porcine embryos below 270 mOsm or above 300 mOsm, by altering the NaCl concentration, decreased development to the blastocyst stage (Figure 2) [73]. Contrarily, Hwang et al. [166] observed that increasing osmolality up to 320 mOsm with NaCl or sucrose did not impair development of IVF-derived embryos but surprisingly decreased transcript abundance of BCL2 associated X protein (*BAX*), a proapoptotic gene, compared to the PZM-3 control. Particular amino acids can also act as osmolytes, including glycine and proline. When 10 mM glycine was added to MU1 porcine embryo culture medium at 260, 275, or 300 mOsm, no difference in development to the blastocyst stage was detected [84]. However, culture with supplemental glycine increased the total cell numbers of blastocyst-stage embryos at 275 mOsm, demonstrating a beneficial effect on embryo quality at this osmolality [84].

## 7. Maternal and Paternal Factors Influencing Development

### 7.1. Gilt-Versus Sow-Derived Oocytes

The production of pig embryos in vitro starts with oocytes from either prepubertal gilts or sexually mature sows. Collection of ovaries at the abattoir is dependent upon the type of facility. Some plants slaughter only gilts while others focus on sows and sausage production. It is generally accepted that in vitro-matured oocytes derived from sows are more developmentally competent than oocytes derived from prepubertal gilts [10,167,168,169]. The diameter of the oocyte, thickness of the zona pellucida, and the perivitelline space were found to be larger in sow oocytes compared in gilt oocytes [170]. However, culture conditions can be modified to improve the quality of in vitro-matured oocytes derived from prepubertal gilts to a level similar to oocytes derived from sexually mature sows. For example, addition of fibroblast growth factor 2, leukemia inhibitory factor, and insulin like growth factor 1 (termed FLI) to the maturation medium quadrupled the production of offspring on a per oocyte basis [11]. Indeed, in the presence of FLI, gonadotropins are unnecessary during oocyte maturation [25]. 

### 7.2. Sperm Quality

Sperm quality is defined in three major categories: sperm motility, sperm morphology, and sperm biomarker-defined health. Sperm motility can be objectively measured using computer-assisted sperm analysis (CASA) systems [171]. These are typically phase-contrast microscopes fitted with cameras and attached to a computer to perform motility calculations. Software on these computers can likewise perform sperm morphology assessment. Common sperm morphology abnormalities include retained cytoplasmic droplets (proximal and distal in relationship to the sperm midpiece), distal midpiece refluxes, and coiled tails. 

There has been increasing attention to evaluating sperm health and function from a biomarker perspective. The advantage to evaluating sperm with biomarkers is that it can reveal sperm quality characteristics that cannot be evaluated under optical microscopes (conventional light microscopy) alone. Biomarkers can be assessed by using microscopes equipped with epifluorescence or through flow cytometry. While there are several characteristics that can be analyzed, we will mention those that are relevant along with manufacturer in parentheses. A common sperm health attribute to assess is acrosomal integrity. Fluorescent conjugated lectins can be used for this. In pig, a common lectin to assess acrosome integrity is Arachis hypogaea/peanut agglutinin (Invitrogen Lectin PNA, Alexa Fluor™ 488) [172]. Plasma membrane integrity can give insight to sperm viability [173] and/or if there has been capacitation-associated plasma membrane remodeling [174]. This is commonly analyzed using propidium iodide (Invitrogen). Mitochondrial membrane potential can be assessed using JC-1 (Invitrogen) [175], while capacitation indicating calcium influx status can be evaluated using Fluo 4 NW (Invitrogen) [176]. An additional capacitation status indicator is the efflux of zinc, which can be analyzed using FluoZin™-3 AM (Invitrogen) [174]. Misfolded sperm protein content can be evaluated for with PROTEOSTAT aggresome kit (Enzo Life Sciences) [177] along with sperm proteins marked for recycling via the ubiquitin proteasome system pathway with antibodies against ubiquitin [178]. Reactive oxygen species can be indicated by 2′,7′-dichlorodihydrofluorescein diacetate (H2DCFDA, Invitrogen) [179] or membrane lipid peroxidation from ROS by 4, 4-difluoro-5-(4-phenyl-1,3-butadienyl)-4-bora-3a,4a-diaza-s-indacene-3-undecanoic acid, C11-BODIPY (BODIPY, Invitrogen) [180]. Lastly, DNA fragmentation can be evaluated by acridine orange (Acros Organics) [181] or TUNEL (Promega DeadEnd™ Colorimetric TUNEL System) [182].

While it is assumed that lower quality sperm can be used for successful fertilization in IVF compared to artificial insemination, sperm quality is still important. To emphasize this, recent studies have shown that flow cytometer (FC)-sorted sperm result in reduced IVF rates [183]. It is well understood that FC-sorted sperm have reduced sperm quality [184,185]. Even though sperm may undergo successful penetration of the zona pellucida, a series of events are necessary to activate the oocyte. Two sperm-oocyte activating factors implicated in activating the oocyte are phospholipase C zeta (PLCζ) [186] and postacrosomal WW domain-binding protein (PAWP) [187]. After sperm-oocyte activation, billions of zinc ions are released, termed the zinc spark (Xenopus [188], cattle [189], and mice [190]). While the zinc spark has not been reported in pigs, we previously reported that the zinc chelator, TPEN, activates pig oocytes [191]. Recent data have shown that zinc inhibits sperm zona pellucida proteinases, such as matrix metalloproteinase-2 (MMP2), and the 26S proteasome [192]. Altogether, sperm-oocyte zinc signaling might be a new polyspermy defense mechanism [174,193].

### 7.3. Epigenetics and In Vitro Production of Porcine Embryos

Creating or culturing pig embryos in vitro changes the developmental potential [112] as well as the transcriptional [12] and epigenetic profile [194,195,196] of the resulting embryos. Some would argue that the change in transcription is due to changes in DNA methylation. While there are reports of epigenetic control over transcription, it is not clear if the changes in DNA methylation, for example, are the result of or the cause of changes in transcription [197,198,199,200]. In pigs, epigenetic regulation of development is not limited to DNA methylation and includes histone modifications, such as acetylation [201], methylation [202,203], and RNA methylation [204]. Much work has been reported on epigenetic regulation in pig development and after SCNT [205,206]. Increasing histone acetylation by inhibiting deacetylase inhibitors improves the development of SCNT embryos and corrects the expression of some genes [201,207,208]. Presumably changes in the culture environment can bring the transcriptional profile and epigenetic marks, such as DNA methylation and histone acetylation, to a more ‘in vivo’ level. 

## 8. Conclusions and Perspectives

Progress in the in vitro production of porcine embryos has been steady; however, with advances in technologies, such as deep sequencing and metabolomics, the conditions required for healthy oocytes and embryos will be revealed more rapidly. In turn, improvements in media formulations and implementation of dynamic culture systems have great potential for dramatically increasing developmental outcomes. Nevertheless, environmental stressors and parental factors that cannot be controlled will continuously pose challenges for the culture of porcine embryos. These challenges will be faced directly to improve embryo viability as porcine models are becoming more important and applicable in agriculture and human medicine.

## Figures and Tables

**Figure 1 cells-10-02770-f001:**
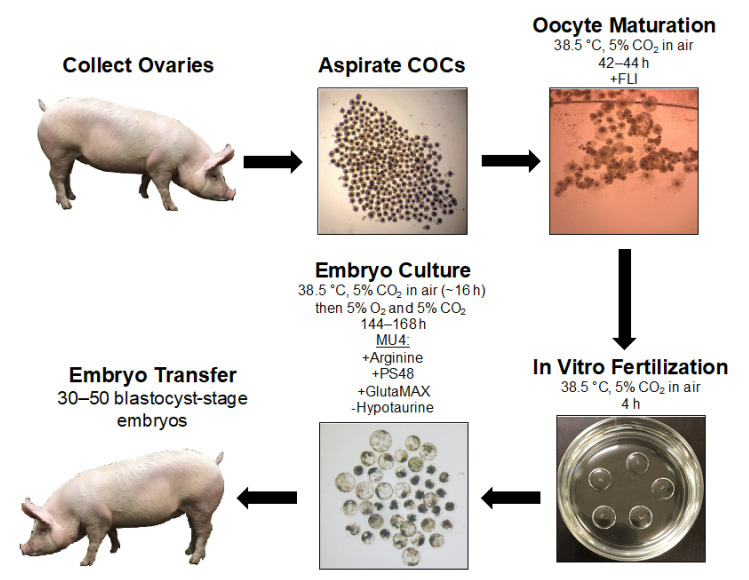
An overview of our current porcine embryo production pipeline. FLI: cocktail of fibroblast growth factor 2, leukemia inhibitory factor, and insulin-like growth factor 1. MU4 porcine embryo culture medium is a modified porcine zygote medium (PZM) that contains supplemental arginine, PS48, and GlutaMAX, and hypotaurine is removed from the formulation. Approximately 30–50 day 5 or 6 morula- and blastocyst-stage embryos are surgically transferred into a surrogate.

**Figure 2 cells-10-02770-f002:**
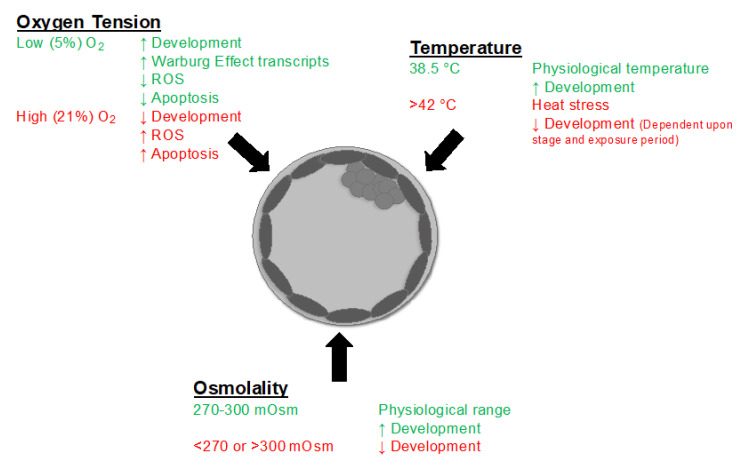
A summary of environmental stressors on porcine embryo development in vitro. Positive conditions are in green, and negative conditions are in red.

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
