# Peer review of "Challenges and Considerations during In Vitro Production of Porcine Embryos"

_cells, 2021, doi:10.3390/cells10102770_

Round 1

Reviewer 1 Report

Review the manuscript Cells 1412012

After review I have some comments, observations and suggestions.

It is very interesting article.

The main elements and methodologies of oocytes maturation, fertilization and embryo development to increase the production and quality of blastocysts produced in vitro are described in detail. All of this in order to obtain good quality offspring for porcine production, besides to be used as a research model for assisted reproduction.

The article is very comprehensive, then  will be very helpful to readersfrom several fieldsin assisted reproduction in humans.

Suggestions:

It could be important to add a sectionabout the use of sperm cryopreservation and embryos vitrification.

There are recent pulished papers about the use of ICSI and PICSI to increase the number and quality of blastocyst. These information could be included.

SPECIFIC OBSERVATIONS

I suggest "in vitro" should be in italics along the manuscript.

Write the abbreviation"IVF"after the full word (line 108)and use it along the manuscript. Lines 130, 291, 368, 373.

Image of COCs showing the cumulus cells expansion during the oocytes maturation should be improved.

Author Response

Thank you for your comments and suggestions. We have revised the manuscript accordingly. Please find our responses below.

It could be important to add a section about the use of sperm cryopreservation and embryos vitrification.

Response: A section on cryopreservation was added from lines 473-518.

There are recent published papers about the use of ICSI and PICSI to increase the number and quality of blastocyst. These information could be included.

Response: Information on ICSI and PICSI was added from lines 145-159.

SPECIFIC OBSERVATIONS

I suggest "in vitro" should be in italics along the manuscript.

Response: This has been changed throughout the manuscript.

Write the abbreviation "IVF" after the full word (line 108) and use it along the manuscript. Lines 130, 291, 368, 373.

Response: This has been changed throughout the manuscript.

Image of COCs showing the cumulus cells expansion during the oocytes maturation should be improved.

Response: A new picture of cumulus cell expansion was added to Figure 1.

Reviewer 2 Report

This manuscript reviews current protocols and procedures for in vitro oocyte maturation, fertilization, and embryo culture in swine. Special emphasis has been placed on  metabolomic aspects related with porcine embryo in vitro culture, as well as on the usefulness of new technologies for the improvement of the current in vitro culture system.

The authors have conducted a very elegant review with a lot of very important and relevant information supported by a deep knowledge of the literature on this topic together with a deep understanding of in vivo and in vitro porcine embryonic development. The manuscript is very well organized and easy to follow. The content of the article is, in my opinion, very interesting since the authors proposed several strategies that could contribute to optimizing the results of in vitro embryo culture systems in pigs. All the efforts made in this regard are highly relevant today. The authors have produced a very good review manuscript, describing the different relevant aspects involved in the in vitro production of porcine embryos.

Based on all of the above, my suggestion is to accept this manuscript.

Author Response

Thank you for your kind comments regarding our manuscript.

Reviewer 3 Report

See extra file General comments pig IVF. I could not paste my report in an orderly manner

Author Response

Thank you for your helpful comments and suggestions. We have revised the manuscript accordingly. Please find our responses below.

Specific comments

Line 79 -85 : “General selection parameters to determine oocyte competence in the pig are the age of the oocyte donor [17,18], follicle size [19], and the layers of cumulus cells and subsequent cumulus cell expansion during culture [20]. Oocytes that originate from small (<3 mm) follicles are less likely to mature to metaphase II (MII) or develop to the blastocyst stage compared to oocytes derived from medium (3-5 mm) and large (>5 mm) follicles [21]. Historically, cumulus cell expansion has been considered to be required for successful maturation in vitro and often the degree of cumulus cell expansion is correlated with improved oocyte maturation [22] [23].”

Please check the recent publications of the Wageningen group on this topic https://pubmed.ncbi.nlm.nih.gov/?term=costermans+n+kemp&sort=date

They found some interesting links between FF steroid profile and COC quality in sows, and on the negative energy balance on oocyte quality.

Response: Two citations from this group have been added to the Oocyte Quality and Maturation section from lines 84-86 and 89-92.

Line 346-348: There are more recent data on chromosomal aberrations in bovine embryos with more powerful techniques than FISH, which was indeed commonly used 20 year ago (116. Viuff, D.; Rickords, L.; Offenberg, H.; Hyttel, P.; Avery, B.; Greve, T.; Olsaker, I.; Williams, J.L.; Callesen, H.; Thomsen, P.D. A high proportion of bovine blastocysts produced in vitro are mixoploid. Biol Reprod 1999, 60, 1273-1278, doi:10.1095/bi- 904 olreprod60.6.1273.)

Please check out Tsuiko et al. 2017 Hum Reprod. 2017 Nov 1;32(11):2348-2357. doi: 10.1093/humrep/dex286. and Destouni et al. 2016 Genome Res 2016 May;26(5):567-78. doi: 10.1101/gr.200527.115. and add those references to the review

Response: These citations were added at lines 371-374.

Line 349 – 350 : it is a bit awkward to refer here at once to the rhesus monkey – data on chromosomal errors in human embryos were acquired already before that, and are probably more important, see : Vanneste et al. (2009) Chromosome instability is common in human cleavage-stage embryos. Nat Med 2009 May;15(5):577-83. doi: 10.1038/nm.1924.

Response: We decided to delete the reference to the rhesus monkey as we agree that this deviated from the focus of the review.